# Structural Features and Physiological Associations of Human 14-3-3ζ Pseudogenes

**DOI:** 10.3390/genes15040399

**Published:** 2024-03-24

**Authors:** Haroon Lughmani, Hrushil Patel, Ritu Chakravarti

**Affiliations:** Department of Physiology and Pharmacology, The University of Toledo, Toledo, OH 43614, USA; haroon.lughmani@rockets.utoledo.edu (H.L.); hrushil.patel@rockets.utoledo.edu (H.P.)

**Keywords:** YWHAZ, pseudogenes, 14-3-3ζ, Alphafold, behavioral disorders, inflammatory arthritis, autoimmune diseases, immune disorders, 14-3-3

## Abstract

There are about 14,000 pseudogenes that are mutated or truncated sequences resembling functional parent genes. About two-thirds of pseudogenes are processed, while others are duplicated. Although initially thought dead, emerging studies indicate they have functional and regulatory roles. We study 14-3-3ζ, an adaptor protein that regulates cytokine signaling and inflammatory diseases, including rheumatoid arthritis, cancer, and neurological disorders. To understand how 14-3-3ζ (gene symbol YWHAZ) performs diverse functions, we examined the human genome and identified nine YWHAZ pseudogenes spread across many chromosomes. Unlike the 32 kb exon-to-exon sequence in YWHAZ, all pseudogenes are much shorter and lack introns. Out of six, four YWHAZ exons are highly conserved, but the untranslated region (UTR) shows significant diversity. The putative amino acid sequence of pseudogenes is 78–97% homologous, resulting in striking structural similarities with the parent protein. The OMIM and Decipher database searches revealed chromosomal loci containing pseudogenes are associated with human diseases that overlap with the parent gene. To the best of our knowledge, this is the first report on pseudogenes of the 14-3-3 family protein and their implications for human health. This bioinformatics-based study introduces a new insight into the complexity of 14-3-3ζ’s functions in biology.

## 1. Introduction

Recent advances in genetics are challenging the previous notations of the modification and regulation of one gene and providing us with new tools to evaluate complex diseases. Pseudogenes are genomic DNA sequences that resemble mutated or truncated versions of a known functional gene [1]. With 10% of genes generating pseudogenes, the human genome has 12,000–20,000 pseudogenes [2]. Most pseudogenes are classified into two classes: processed and duplicated. Processed pseudogenes account for almost 2/3 of all pseudogenes and originate using long interspersed element-1 (LINE-1) retrotransposons to reverse transcribe gene mRNA and reinsert it into other genome regions. The LINE-1 retrotransposon produces reverse transcriptase and endonucleases to create cDNA from the mRNA and re-insert the cDNA throughout the genome. These pseudogenes lack introns and closely resemble the mRNA sequence of the parent gene. On the other hand, duplicated pseudogenes include introns and other regulatory elements from the parent gene [3].

Historically, it was long presumed that pseudogenes, lacking the extensive regulatory framework of normal genes, cannot be transcribed or translated. Despite lacking a complete regulatory framework, studies indicate that pseudogenes with features such as open chromatin, TSS histone marks, Pol2 binding sites, and transcription factor binding sites may still be active [4]. Recent studies suggest that up to 10% of pseudogenes are transcribed, and 40% of transcripts are translated into protein [5,6,7]. The pseudogene transcripts can act as IncRNA or antisense RNA, thus regulating parent gene expression, e.g., *HMGA1*, *OCT4*, and *PTEN* [8]. Further evidence suggests that pseudogene-derived proteins can affect signal transduction and participate in human diseases [9]. Examples include the pseudogene of UBB (UBBP4), which translates and ubiquitinates proteins, leading to the nuclear accumulation of Lamins or HBBP1 with implications for human erythropoiesis [9,10].

We are interested in understanding the role of 14-3-3ζ in immune dysfunctions. 14-3-3ζ is a highly conserved and universally present protein, and its adaptor function allows it to interact with hundreds of proteins and participate in several signaling pathways, notably the RAS-RAF pathway and IL-17A signal transduction [11,12]. 14-3-3ζ-deficient animals, although viable, are sterile, have neurological and cognitive disorders, and are susceptible to inflammatory arthritis [13,14,15]. Several transcriptomics and specific case studies have eluded the association of 14-3-3ζ with several human diseases, including diabetes, Alzheimer’s disease, schizophrenia, rheumatoid arthritis, and viral infections [15,16,17,18,19,20,21]. To understand how 14-3-3ζ can participate and regulate various phenotypes affecting various tissues and cell types, we examined its origin at the genomic level using bioinformatics tools and recently released datasets [22,23]. In addition to the parent YWHAZ gene, we noticed the presence of nine YWHAZ pseudogenes of varied lengths on different chromosomes and with distinct tissue expression profiles. Intrigued by it, we performed a detailed analysis of their exons, cDNA composition, putative amino acid sequence, and hypothetical protein structure, with a prediction of their physiological relevance. Our results indicate that there is a significant sequence and structure conservation between YWHAZ and pseudogenes that extends to a commonality of physiologically relevant musculoskeletal and neurological disorders, suggesting that the pseudogenes may have a role in supporting 14-3-3ζ function in physiology.

## 2. Materials and Methods

### 2.1. Genetic Features

The location of the human YWHAZ, its pseudogenes, and its transcripts were determined using the ensembl.org (accessed on 25 February 2024) GRCh38.p14 genome assembly (GenBank Chromosome Assembly: GCA_000001405.29) [24]. The cDNA sequences used were located from the Ensembl release 111 in January 2024. The presence of polyadenylation sequences on the pseudogenes and their PAS scores were identified using the APARENT PolyA Detector from the University of Washington [25]. The presence of kozak sequences in the pseudogene DNA and the initiation codons were determined using the ATGpr program [26]. The Sequence Manipulation Suite was used to determine the number of CpG islands per pseudogene [27].

### 2.2. Sequence Alignment and Analysis

The DNA nucleotide alignment, DNA exon conservation alignment, cDNA alignment, and amino acid alignment of YWHAZ and pseudogenes were determined using the CLUSTALW Multiple Sequence Alignment [28]. These alignments were then further analyzed using MView to display differences clearly and to create consensus sequences [29]. The NCBI ORF finder was used to determine the amino acid sequence of pseudogenes from the cDNA sequence. After filtering through the potential results, the sequence with the highest conservation to the 14-3-3ζ parent protein was selected and used for sequence comparison [30]. Amino acid sequence analysis was performed using the CLUSTALW Multiple Sequence Alignment tool, which was compiled and manually marked at sequence mismatch locations to the parent protein [28,30,31]. The degree of conservation of amino acids was also demonstrated using WebLogo [32].

#### Genome Browsers

Vega: https://vega.archive.ensembl.org/Homo_sapiens/Search/Results?q=ywhaz;site=vega;facet_feature_type=;facet_species=Human;perpage=50 (accessed on 25 February 2024)GTEX: https://gtexportal.org/home/gene/YWHAZ, https://gtexportal.org/home/gene/YWHAE (accessed on 25 February 2024)ENSEMBL: https://useast.ensembl.org/Human/Search/Results?q=ywhazp;site=ensembl;facet_species=Human (accessed on 25 February 2024)UCSC: https://genome.ucsc.edu/cgi-bin/hgTracks?db=hg38&lastVirtModeType=default&lastVirtModeExtraState=&virtModeType=default&virtMode=0&nonVirtPosition=&position=chr8%3A100916523%2D100952020&hgsid=2046927750_ilbAM0OszG5ESmnTET9MfaDrmaLN (accessed on 25 February 2024)Consensus CDS Protein Set: https://www.ncbi.nlm.nih.gov/CCDS/CcdsBrowse.cgi?REQUEST=CCDS&DATA=CCDS6290.1 (accessed on 25 February 2024)EBI Expression Atlas: https://www.ebi.ac.uk/gxa/experiments/E-MTAB-2836 (accessed on 25 February 2024)

### 2.3. Expression Data

Tissue-specific expression and relative levels (TPM) of data for each pseudogene was derived from RNA-seq baseline experiments from the Genotype-Tissue Expression (GTEx) project (Accession Number: phs000424.v8.p2) [22]. The expression data for YWHAZ and YWHAE pseudogenes were compiled into a heatmap. Bone marrow-specific expression of several YWHAZ pseudogenes was obtained from the expression atlas (E-MTAB).

### 2.4. Structure Simulation

AlphaFold2 was used to simulate the structure of potential pseudogene-derived proteins using the putative amino acid sequences, determined using the NCBI ORF Finder [33] (https://colab.research.google.com/drive/1wRo0HB61bQ1NSbRxXEnd7Te3pSC4RyCg?usp=sharing, accessed on 25 February 2024).

SWISS-MODEL was used to simulate pseudogene protein structure using the reported crystal structure of human 14-3-3ζ (SMTL ID: 7q16.1) as a template along with the putative amino acid sequences [34]. The SWISS-MODEL software was used to overlap the predicted pseudogene protein structure on top of the WT 14-3-3ζ protein structure.

### 2.5. Association with Human Disease

The OMIM (Accession Numbers: RCV000778077.6, RCV000778076.6, RCV000778075.6, and RCV000778074.6), Decipher (https://www.deciphergenomics.org/gene/YWHAZ/overview/clinical-info, accessed on 25 February 2024) and *ClinVar* databases (Accession Numbers: VCV000448894.13, VCV000151955.1), along with the published literature, were used to scan YWHAZ and its clinical variants associated with human disease [30,35]. The association of pseudogenes with human disease was determined by searching the DECIPHER database and individual loci-associated scientific literature [36].

## 3. Results

### 3.1. YWHAZ Pseudogenes in Human Genome

The parent YWHAZ gene structure is present on the human chromosome 8q and spans about 32kb region (Figure 1a, Table 1). The GRCh38.p14 assembly shows 23 transcripts of the YWHAZ gene spanning 567 to 5248 nucleotides in length. While all transcripts contain introns and exons, eighteen transcripts code proteins with a size of 51–245 amino acids (Appendix A). In addition to the transcripts, we noticed several pseudogenes in the nucleotide blast search of the YWHAZ gene. There was no published study of YWHAZ pseudogenes, which encouraged us to compile the first detailed information report about these. The Vega archive for YWHAZ within the human genome lists fourteen pseudogenes, Genotype-Tissue Expression (GTEx) lists seven, and *Ensembl* lists nine pseudogenes on various chromosomes. In contrast to 14-3-3ζ, other 14-3-3 isoforms of the seven-membered family show none (14-3-3γ (YWHAG) and 14-3-3η (YWHAH)) to few (14-3-3ε (YWHAE) and 14-3-3θ (YWHAQ) have 6 and 7, respectively, according to Ensembl’s release 111) pseudogenes [37]. Most YWHAZ pseudogenes are located on separate chromosomes, except two on number ten and three on X. Notably, no pseudogene exists on chromosome #8, where the parent YWHAZ gene is located (Table 1). The approximate length of pseudogenes is around 730 base pairs (bp), ranging from 631 to 1034 bp for YWHAZP7 and YWHAZP4, respectively (Figure 1a,b). The overall nucleotide sequence similarity of pseudogenes with YWHAZ ranges from 19 to 42%; however, higher conservation is observed within the coding region. Hence, the cDNA conservation averages much higher at 92.05%, ranging between 80.6% and 98% (Table 2 and Appendix A).

### 3.2. Characteristics of Pseudogenes

The standard transcript of YWHAZ is around 5.2 Kb, which spans over six exons interspersed with five introns. All pseudogenes, except for YWHAZP4, have only one exon. Transcripts of most pseudogenes are around 730 bp (Table 1). As indicated by the much shorter DNA lengths and highly conserved cDNA, most YWHAZ pseudogenes are derived from mRNA and thus belong to the processed pseudogene category. However, unlike the original ‘one exon-no intron’ rigid processed pseudogene theory, YWHAZP4 contains two exons with an intron region between them, resulting in a longer transcript of 1034 bp (Figure 1a). YWHAZP4 also has a 5′ promoter, enhancer, and CTCF region. This indicates that YWHAZP4 may belong to the duplicated pseudogene category and, like genes, can have a complex gene structure. However, both Vega and Ensembl list YWHAZP4 as a processed pseudogene. Out of six, four exons (2, 3, 4, and 5) of YWHAZ are significantly conserved, while exons #1 and #6 show significant diversity in all pseudogenes (Figure 1c). YWHAZP5 has the least sequence similarity of all pseudogenes, and YWHAZP4 is the most conserved at the genomic level (Table 2). The increased transcript length and intron sequence overlap may account for increased sequence similarity for YWHAZP4.

It is reported that 40% of lncRNA and pseudogenes are translated into peptides of >10aa in length in vivo [7]. We examined if the YWHAZ pseudogenes transcripts carry the standard signatures of translation. All, except YWHAZP1, pseudogenes transcripts have initiation codon. The universal initiation codon (AUG) is present in all the transcripts except YWHAZP7 and YWHAZP8, where only the alternate codon TTG is present. While 5′ UTR and 3′UTR were found in most pseudogenes, the Kozak sequence is present in 6 out of 9 genes (Table 2) [26]. Finally, the PAS score shows the presence of a Poly-A tail for most pseudogenes (Table 2 and Appendix A) [38].

### 3.3. Putative Pseudogene-Derived Proteins and 14-3-3ζ

Based on the open reading frame (ORF) that best modeled the sequence of 14-3-3ζ, the putative amino acid sequences for all pseudogenes were determined from the cDNA (Appendix A) [30]. The sequence alignment shows significant regions of similarities and differences among the pseudogenes. Most pseudogenes show sequence similarity with the parent protein between residues 20 and 180 (Appendix A). Among the group, YWHAZP3 and YWHAZP10 appear most like YWHAZ with nearly complete conservation (Figure 2a). We used WebLogo to illustrate the amino acid conservation of pseudogene-derived proteins with the 14-3-3ζ [32]. While methionine remains the first amino acid in all proteins, many amino acids are conserved in pseudogenes. Notably, the N-terminal is more conserved than the C-terminal region; particularly, conservation is reduced after Ser^175^ (Figure 2b). However, the overall amino acid alignment score for pseudogenes ranged from 80.4–97.6%, indicating a high level of sequence conservation with 14-3-3ζ (Table 2). To further understand the three-dimensional structure, we simulated the structures of putative pseudo proteins using AlphaFold 2 and observed secondary structures primarily consisting of α helices [33]. We next compared the structures with parent 14-3-3ζ using SWISS-MODEL and generated overlay models [34]. While each putative protein differs in length, the overlay on the 14-3-3ζ showed almost complete conservation of the helical structure with minor changes in the disordered regions. Notably, mutations or variations in pseudogene sequence do not significantly affect the structure. Importantly, structures generated by AlpaFold2 and SWISS-MODEL were quite similar (Figure 3 and Appendix A).

### 3.4. Expression of Pseudogenes

We used the GTEx database to determine the expression of the pseudogene transcripts and compare it with the parent YWHAZ [22]. First, we noticed YWHAZP2, YWHAZP3, YWHAZP4, YWHAZP5, YWHAZP6, YWHAZP7, and YWHAZP8 (limited expression data are available for YWHAZP1 and YWHAZP10) are expressed at much lower levels than the parent gene. Second, amongst pseudogenes, YWHAZP3, YWHAZP4, and YWHAZP5 are expressed at higher levels than others. Third, specific tissues, such as the reproductive organs, brain, and mucosa, show higher levels of pseudogene transcripts, while many other tissues show little to no expression (Figure 4). To understand the specificity of 14-3-3ζ, we compared the heatmap for the expression of 14-3-3ε’pseudogenes, which has several differences in the expression level and sites (Appendix A), thus indicating the specificity of pseudogene expression.

We compared primary expression sites for the parent YWHAZ transcript to understand whether the tissue-specific expression is relevant. The top five expression sites for YWHAZ are the esophagus mucosa, vagina, brain, EBV-transformed lymphocytes, and non-sun-exposed suprapubic skin. The top five expression sites for most pseudogenes transcript overlap with YWHAZ; primarily, the vagina, mucosa, lymphocytes, and brain are common ones. Only a few pseudogenes are expressed in arteries and lymph nodes (Figure 5a). At the cellular level, the current GTEx dataset includes only two cell types—fibroblasts and EBV-transformed lymphocytes. The expression level of 14-3-3ζ in both cells has functional consequences affecting disease outcomes [11,39,40]. When the relative expression of YWHAZ and pseudogenes is compared, the YWHAZ transcript is expressed at a high level in both fibroblasts and lymphocytes, while pseudogenes are expressed at much lower levels. Only YWHAZP2, YWHAZP3, YWHAZP4, and YWHAZP5 exceed the 0.1 transcripts per million (TPM), an accepted criterion for pseudogene expression [41] (Figure 5b).

### 3.5. Physiological Aspect of Pseudogenes

At the genetic level, the 14-3-3ζ (YWHAZ) gene is associated with human developmental delays, intellectual disability, epilepsy, dysmorphic features, amyotrophic lateral sclerosis, and heart diseases [19,40,42,43,44,45,46] (Table 3). Genetic and transcriptomics studies show changes in 14-3-3ζ protein expression with several human diseases, including musculoskeletal, inflammation, and cancer [19,42,43,47]. Similarly, YWHAZ deficiency in animals results in developmental delays, increased inflammation, and musculoskeletal deficits [13,45,48].

To understand the relevance of YWHAZ pseudogenes, we used the following two approaches: First, we searched for disease association with chromosomal regions containing YWHAZ pseudogenes. Similar to the YWHAZ phenotype, the chromosomal regions harboring YWHAZP4 (6q22.33), YWHAZP5 (10q25.1), YWHAZP6 (9p13.3), YWHAZP8 (Xq13.2), and YWHAZP10 (Xp11.4) are associated with intellectual disability [49,50,51,52,53]. Several case studies indicate deletions in the chromosomal region containing YWHAZP4 (6q22.33) is linked to intellectual disability [49]. The chromosomal band containing YWHAZP5 has been linked to Coffin–Lowry syndrome, which presents as severe intellectual disability and physical deformities [50]. Interestingly, a copy number gain variant in the specific region of the YWHAZP6 (9p13.3) gene has been associated with developmental and intellectual disabilities in a clinical case study (ClinVar Accession: VCV000151955.1). Duplication of Xq13.2-Xq21.31, the region containing YWHAZP8, along with other genes such as ATRX and PCDH11X, has been reported in a clinical case associated with X-inactivation, resulting in developmental delays and seizures in a boy with pubertal gynecomastia [51]. Furthermore, the YWHAZP10 gene is located next to OTC and CASK, the deletion of which results in hyperammonemia and X-linked intellectual disability, respectively [52,53]. While not explicitly related to intellectual disability, the YWHAZP1-containing chromosomal band is associated with migraines [54] (Table 3). An in-depth study of the DECIPHER database further showed that both duplication and deletion of the most loci containing YWHAZ pseudogenes are associated with abnormal human phenotypes. If correct, then it would suggest that regulation of pseudogene expression. It is important to note that multiple genes are located in that chromosomal region that shows phenotypic association; therefore, the phenotype can be attributed to a cumulative effect or additional genes of importance present in the region.

Second, we manually mined the transcriptomics-based studies for pseudogenes in human diseases. A recent study shows that YWHAZP5 is one of the 40 biomarkers of adult immune thrombocytopenia in humans [55]. The locus associated with YWHAZP7 is expressed in glioblastomas [56]. Decreased expression of YWHAZP7 and YWHAZP10 in the platelets of six different tumor tissues is also reported [57]. A detailed transcriptomic analysis of the pseudogenes in human cancers shows several YWHAZ pseudogenes that are distinct from the ones reported in (YWHAZP1-YWHAZP10) in Ensembl (Table 4) [58]. These pseudogenes differ in chromosomal locations, suggesting that the total count of YWHAZ pseudogenes in the human genome is not yet final.

We recently reported that YWHAZ-deficient animals show significant bone loss in the inflammatory arthritis model, which can be prevented by immunizing animals with a 14-3-3ζ-based vaccine [15]. Therefore, we searched for the expression of pseudogenes in cell types associated with inflammation and bone remodeling. A study of T cells from the blood and synovial fluid of psoriatic arthritis patients revealed the expression of YWHAZP3 and YWHAZP10 in both leukocytes and T cells; the expression of YWHAZP1, YWHAZP5, and YWHAZP6 in leukocytes only; the expression of YWHAZP2 in T Cells only [59]. Similarly, synovial tissue macrophages from patients in remission from rheumatoid arthritis showed the expression of YWHAZP3, YWHAZP4, and YWHAZP10 [60]. Additionally, YWHAZP2, YWHAZP3, and YWHAZP10 expression is elevated in a study of LPS-treated PBMC of juvenile idiopathic arthritis patients [61]. This encouraged us to examine the cell-specific expression of pseudogenes in the human expression atlas (EMBL-EBI). It was interesting to observe detectable to significant levels of YWHAZP2, YWHAZP3, YWHAZP4, and YWHAZP10 in several hematopoietic cells, including CD4 and CD8 lymphocytes and macrophages [59]. Four pseudogenes, YWHAZP2, YWHAZP3, YWHAZP7, and YWHAZP10, are found to be expressed in the bone marrow. Due to insufficient cellular details, further specificity to osteoblast, chondrocytes, and osteoclast could not be made.

**Table 3 genes-15-00399-t003:** Human diseases associated with the genetic loci of YWHAZ and its pseudogenes.

Gene	Genomic Loci	Diseases	Reference
YWHAZ	8q22.3	Developmental delays, Intellectual disability, dysmorphism, Amyotrophic lateral sclerosis, Attention deficit hyperactivity disorder, Heart function.	[19,39,41,42,44,45,46]
YWHAZP1	14q21.2	Migraines and expression in psoriatic arthritic leukocytes.	[53,58]
YWHAZP2	2q14.3	Expression in Psoriatic Rheumatoid Arthritic T-Cells, elevated expression in LPS-treated PBMC of juvenile idiopathic arthritis patients.	[58,60]
YWHAZP3	10p12.2	Expression in Psoriatic Rheumatoid Arthritis leukocytes and T-Cells as well sa expression in rheumatoid arthritic synovial macrophages. The gene has elevated expression in LPS-treated PBMC of juvenile idiopathic arthritis patients.	[58,59,60]
YWHAZP4	6q22.33	Intellectual Disability, a transcriptomic marker in synovial macrophages for patients in remission with rheumatoid arthritis.	[48,59]
YWHAZP5	10q25.1	Chromosomal region associated with Coffin-Lowry Syndrome, Expression in Psoriatic Rheumatoid Arthritic leukocytes, and 1 of 40 biomarkers for immune thrombocytopenia.	[49,58,62]
YWHAZP6	9p13.3	Expression in Psoriatic arthritic leukocytes, contains a copy number gain variant associated with developmental delay.	[63] ClinVar Accession: VCV000151955.1
YWHAZP7	Xq11.2	Chromosomal region Xq11.1–Xq11.2 found to be amplified in study of glioblastoma cells.	[64]
YWHAZP8	Xq13.2	Xq13.2-q21.31 duplication found in boy with recurrent seizures and pubertal gynecomastia.	[51]
YWHAZP10	Xp11.4	Associated with hyperammonemia and X-linked intellectual disability. Also expressed in synovial macrophages in rheumatoid arthritis and in LPS-treated PBMC of juvenile idiopathic arthritis patients.	[51,52,59,60]

Major disease associations at the genetic loci of YWHAZ and its pseudogenes are listed above. Many of the pseudogenes share physiological phenotypes similar to those of the parent YWHAZ.

**Table 4 genes-15-00399-t004:** YWHAZ pseudogenes identified in the cancer transcriptome.

Chromosomal Loci	Cancer
chrX:41,420,046–41,420,852	Lymphoma, Breast
chr6:127,717,472–127,720,031	Colon, Kidney
chr15:43,132,056–43,133,113	Prostate, Gastric
chrX:41,417,881–41,420,027	Cervical, Lymphoma
chr2:127,030,021–127,032,314	Cervical, Gastric
chr10:107,436,038–107,439,382	Cervical, Colon
chr10:23,465,825–23,468,573	Cervical, Gastric
chr9:34,911,199–34,912,896	Cervical, Lymphoma
chr6:127,720,001–127,722,735	Colon, Prostate
chr6:50,779,065–50,779,689	MPN, Cervical
chr15:43,136,038–43,136,973	Prostate, Melanoma
chr12:17,408,254–17,409,509	MPN, Bladder
chr11:59,601,362–59,604,558	Cervical, Colon
chr15:43,136,038–43,136,973	Prostate, Melanoma
chr12:17,408,254–17,409,509	MPN, Bladder
chr11:59,601,362–59,604,558	Cervical, Colon

Sixteen YWHAZ pseudogenes, identified in a transcriptomic study of cancer cells [58], are listed with their chromosomal locations as well as the cancer specimen they are expressed in. None of these sites match the Ensembl gene locations, indicating the presence of unregistered YWHAZ pseudogenes. Data was adapted from Table S3 of reference [58].

## 4. Discussion

The 14-3-3ζ is an adaptor protein that participates in many cellular signaling pathways and regulates cellular responses and phenotypes [21]. It is also one of the highly conserved mammalian proteins [37]. Compared to the intracellular functions, 14-3-3ζ in the extracellular environment regulates the immune system by its antigenic role and ability to remodel the extracellular matrix [17,65]. Several hypotheses, including the ability to heterodimerize and high abundance, support 14-3-3ζ, performing a wide range of functions [66]. Here, we present that the 14-3-3ζ (YWHAZ) has at least nine pseudogenes on seven different human chromosomes. A decent amount of information for most pseudogenes is available in the Ensembl and GTEx platforms. All pseudogenes have sequences corresponding to coding exons 2–5 while excluding introns; therefore, these pseudogenes can be categorized as processed mRNA. In addition to the coding sequence, YWHAZP4 also contains an intron sequence similar to the original intron between exons 5 and 6. Therefore, YWHAZP4 can be categorized as a duplicate pseudogene; however, both Ensembl and GTEx list it as a processed pseudogene.

Out of six, four (2 through 5) exons are highly conserved in all pseudogenes, and the other two (# 1 and 6) are less conserved with the parent gene. Exons 1 and 6 contain the untranslated regions (UTR) of YWHAZ [23]. Notably, most transcripts have Kozak sequences, an initiation codon, and polyadenylation sequences, suggesting the likelihood of their translation to proteins [25,26] (Table 2 and Appendix A). Although most pseudogenes are expressed at significantly lower levels, the top expression sites show similarities between the YWHAZ and its pseudogenes [22]. Is this indicative of a common promoter or enhancer regulating the transcription of YWHAZ and pseudogenes, remains to be seen. If these pseudogenes are translated, these proteins will likely have a similar structured N-terminal region. The variations observed at the C-terminal end will likely impact binding proteins and may have functional consequences. However, high structural and sequence conservation suggests that these pseudogenes may contribute to the diverse cellular and immunologic functions of YWHAZ.

Originally, pseudogenes were thought to be non-functional and part of lncRNA (long non-coding RNA); however, recent studies indicate that a significant number have promoter activity. Among YWHAZ pseudogenes, YWHAZP4 has promoter, enhancer, and CTCF sites upstream to the gene, blurring the lines between pseudogene and gene. Emerging studies indicate the presence of functional pseudogene proteins for several genes, e.g., PTENP1, cytochrome P450, Notch2NL, UBB, and SRGAP2C [7,8,10]. While there are a limited number of examples of pseudogene-derived proteins, it is well-accepted that pseudogenes are frequently transcribed [2]. Several studies have shown that pseudogenes play an active role in regulating the expression and function of parent genes [63,67]. They can promote or negate the regulatory actions, thereby affecting the expression of the parent transcript, as shown for PTENP1 promotional effect on PTEN transcript levels or the therapeutic effect of gene editing NCF1 pseudogenes on chronic granulomatous disease [8,62]. Many cancer-related studies have reported the increased presence of pseudogene transcripts and their differential expression in a tissue-specific manner [58]. This includes several YWHAZ and its pseudogene transcripts identified in various cancer tissues [58]. Kozak similarity score-based algorithms have shown that non-canonical initiation codons, primarily CTG-based transcripts, are prominently expressed in cancers [58]. It is important to ask if these novel pseudogenes regulate 14-3-3ζ expression or functions in cancer tissues [19]. 

There is a paucity of literature on pseudogenes in autoimmune diseases. Our lab and others have indicated that 14-3-3ζ has antigenic roles in cancer, vasculitis, and rheumatoid arthritis [15,46,47]. Since pseudogenes show high structural conservation, they may also participate in or regulate 14-3-3ζ’s antigenic function. Use of YWHAZP5 as a biomarker of autoimmune thrombocytopenia, increased expression of YWHAZP2, YWHAZP3, and YWHAZP10 in the LPS-stimulated PBMC of systemic juvenile idiopathic arthritis patients, presence of YWHAZP3, YWHAZP4, and YWHAZP10 in synovial tissue macrophages from rheumatoid arthritis patients in disease remission, may suggest their involvement in maintaining immune status [55]. Like YWHAZ, some pseudogenes (YWHAZP4 and YWHAZP6) exhibit tissue-specific expression in the reproductive organs. The tissue-specific skew of YWHAZP4 and YWHAZP6 to reproductive organs corroborates with data that the two highest tissues in which pseudogenes are expressed are the adult testis and fetal ovary [64]. Further investigation is needed to determine if this pseudogene reproductive tissue skew is functionally relevant to the infertile phenotype observed in YWHAZ-knockout animals [13].

Overall, intellectual disability and arthritis emerged as a common phenotypic theme associated with YWHAZ and its pseudogene-harboring chromosomal bands. Since 14-3-3ζ is an essential regulator of RAS-RAF signaling, which plays an important role during development, changes in its expression or function can have severe consequences [43]. The functional relevance of pseudogenes in executing or regulating the phenotype of 14-3-3ζ is an important question. With the ability to have multiple exons and redefine the genetic material, these pseudogenes, like genes, are complex and dynamic, very different from the ‘genetic waste’ they were once considered.

## 5. Conclusions

Since 10% of total genes, primarily essential genes, are responsible for pseudogenes, the presence of several YWHAZ pseudogenes in the human genome is not surprising [9]. However, the spread of pseudogenes on different chromosomes, sequence conservation, and putative structure similarity with the parent YWHAZ is impressive. Though the current study is limited by genetic data and no proteomic evidence related to pseudogenes was present, the common phenotypes associated with chromosomal regions harboring pseudogenes indicate the possibility of functional overlap between the parent and pseudogenes. We expect that this study will highlight the presence of YWHAZ pseudogenes in the human genome. How pseudogenes and their differential expression influence the overall function of the 14-3-3ζ in different diseases remains to be seen. We believe the emerging field of pseudogenes and the role of 14-3-3ζ will help us understand complex human diseases in the near-future.

## Figures and Tables

**Figure 1 genes-15-00399-f001:**
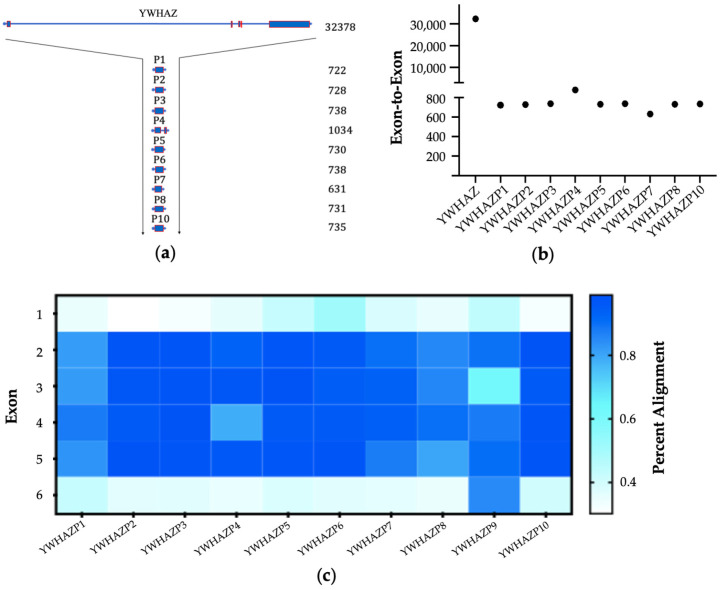
Comparing exons of pseudogenes with parent YWHAZ (**a**,**b**) Schematic of the number of exons (rectangles) in transcript and exon-to-exon length for pseudogenes are compared with YWHAZ; (**c**) Sequence conservation across six exons of YWHAZ is compared for all pseudogenes.

**Figure 2 genes-15-00399-f002:**
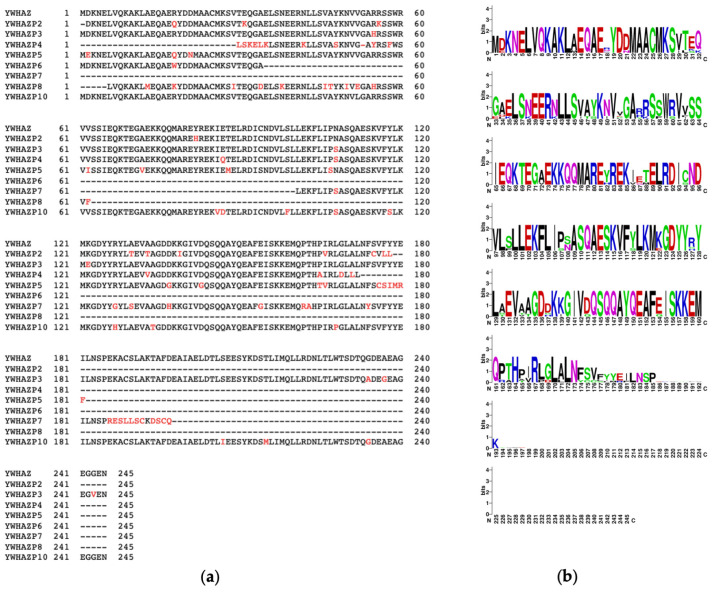
Amino acid conservation between parent YWHAZ and pseudogenes. (**a**) CLUSTALW alignment and (**b**) WebLogo to show conserved amino acids across pseudogenes and parent YWHAZ. Colors and size indicate the type of amino acid and % conservation, respectively.

**Figure 3 genes-15-00399-f003:**
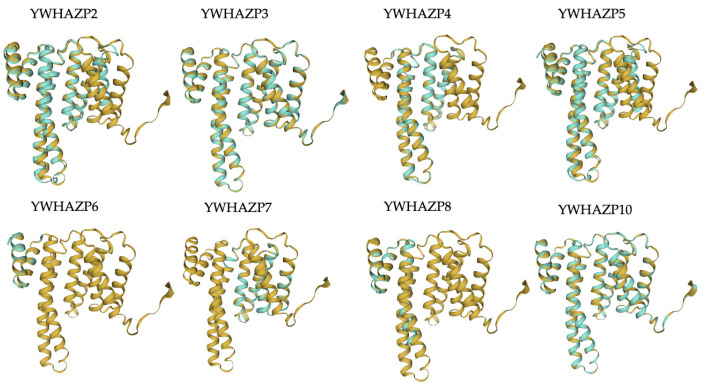
Overlay of parent and putative pseudogene-derived protein structure. The YWHAZ WT protein is shown in gold while putative pseudogene-derived proteins are in turquoise. The overlap shows regions of similarities and differences.

**Figure 4 genes-15-00399-f004:**
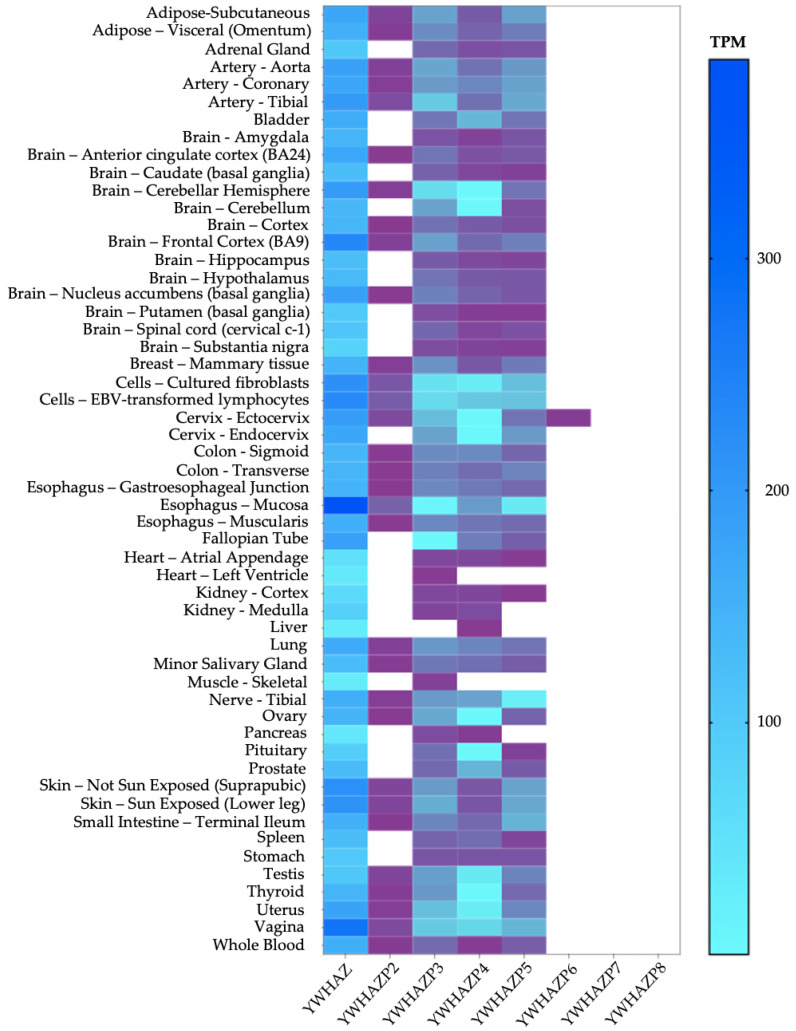
Expression of pseudogenes relative to the original gene in different tissues. The GTEx database was used to obtain a heatmap showing baseline pseudogene expression in various tissues and cells in transcripts per million (TPM). The white color indicates no transcripts are detected; purple indicates low but detectable transcripts in a specific tissue between 0.1 and 1 TPM; blue indicates an increase in transcription from light to dark blue.

**Figure 5 genes-15-00399-f005:**
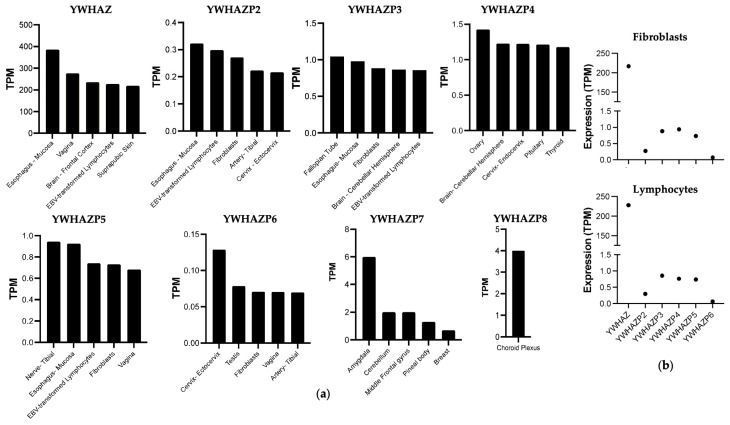
Expression sites of YWHAZ and pseudogenes. (**a**) The topmost five expression sites for each pseudogene and parent YWHAZ gene are shown. (**b**) Relative expression of YWHAZ and pseudogenes in fibroblasts and EBV-transformed lymphocytes is shown.

**Table 1 genes-15-00399-t001:** Location and length of YWHAZ pseudogenes.

Gene	ID	Chromosome	Strand	Locus	Nucleotide Region	DNA Length (bp)	mRNA Length (bp)
YWHAZ	ENSG00000164924	8	Reverse	8q22.3	100,916,523–100,953,388	32,378	5011
YWHAZP1	ENSG00000259148	14	Reverse	14q21.2	44,290,997–44,291,718	722	722
YWHAZP2	ENSG00000213236	2	Reverse	2q14.3	126,557,435–126,558,162	728	728
YWHAZP3	ENSG00000229932	10	Forward	10p12.2	23,136,924–23,137,661	738	738
YWHAZP4	ENSG00000213131	6	Forward	6q22.33	127,355,756–127,356,789	1034	708
YWHAZP5	ENSG00000213260	10	Forward	10q25.1	105,686,322–105,687,051	730	730
YWHAZP6	ENSG00000215199	9	Reverse	9p13.3	34,922,184–34,922,921	738	738
YWHAZP7	ENSG00000227261	X	Reverse	Xq11.2	64,612,632–64,613,262	631	631
YWHAZP8	ENSG00000225664	X	Forward	Xq13.2	73,274,785–73,275,515	731	731
YWHAZP10	ENSG00000217624	X	Reverse	Xp11.4	41,675,760–41,676,494	735	735

The YWHAZ transcript was BLASTed against the human genome to identify the pseudogenes and their locations, as listed above. The DNA length was determined manually from the first exon’s start to the last exon’s end. The mRNA length was determined by matching it with the length of the cDNA released in GRCh38.p14.

**Table 2 genes-15-00399-t002:** Genetic features of YWHAZ pseudogenes.

Gene	DNA Alignment (%)	cDNA Alignment (%)	Amino Acid Alignment (%)	PolyA Tail	Kozak Sequence	CpG Island (#)	Exons	Initiation Codon	Nature of Pseudogene
YWHAZ			100	YES	GXXATGG	3176	6	ATG	
YWHAZP1	19	80.6		YES			1		Processed
YWHAZP2	32	97.1	93.8	YES	GXXATGG	62	1	ATG	Processed
YWHAZP3	31	97.6	97.6	YES	GXXATGG		1	ATG	Processed
YWHAZP4	42	90.5	86.7	YES	CXXATGG		2	CTG/ATG	Processed
YWHAZP5	15	96	90.1	NO	GXXATGG		1	ATG	Processed
YWHAZP6	32	95.1	97.1	NO	GXXATGG		1	ATG	Processed
YWHAZP7	24	88.4	80.4	YES			1	TTG	Processed
YWHAZP8	28	85.2	80.7	YES			1	TTG	Processed
YWHAZP10	32	98.0	95.5	YES	GXXATGG	195	1	ATG	Processed

Genetic features, including the sequence conservation of DNA, cDNA, and amino acids, the presence of a Poly A tail, the Kozak Sequence, and the initiation codon of each pseudogene, are listed.

## Data Availability

All results are accessible and have been shared. The datasets analyzed in this current study are available in the GTEX V8 release (Accession Number: phs000424.v8.p2), OMIM (Accession Numbers: RCV000778077.6, RCV000778076.6, RCV000778075.6, RCV000778074.6), Ensembl (Accession Number: GCA_000001405.28), and NCBI ClinVar repositories (Accession Numbers: VCV000448894.13, VCV000151955.1). All datasets used are cited in the methods section. The 3D pseudogene protein models on SWISS-MODEL were formed based on the 14-3-3ζ template (SMTL ID: 7q16.1.A).

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
