# Peer review of "Structural Features and Physiological Associations of Human 14-3-3ζ Pseudogenes"

_genes, 2024, doi:10.3390/genes15040399_

Round 1
Reviewer 1 Report
Comments and Suggestions for Authors
In the manuscript "Structural Features and Physiological Associations of Human 14-3-3ζ Pseudogenes", Lughmani et al investigated the characteristics and potential physiological roles of YWHAZ pseudogenes in the human genome. The 14-3-3ζ is a highly conserved and universally present protein, and its adaptor function allows it to interact with hundreds of proteins, participate in several signaling pathways, and regulate cytokine signaling and inflammatory diseases, including rheumatoid arthritis, cancer, and neurological disorders. Although lacking experimental data, this manuscript provided a bioinformatics analysis of these pseudogenes' structure, sequence conservation, and their possible links to various human diseases. The study employed bioinformatics tools to analyze pseudogene sequences and their expression across different tissues, suggesting that these pseudogenes may play significant roles in physiological processes. The manuscript is well-structured and presents the information in a logical sequence. Minor grammatical or typographical errors could be addressed for clarity. In addition, this manuscript is not written as in mdpi format, including using different references styles. Figures and Tables were not structured. The methods used for genetic feature analysis, sequence alignment, and structure simulation were detailed and appropriate for the study's aims. However, it would benefit to explain how these methodologies can be associated to contribute to the novelty and conclusions of the study. Indeed, this study provided novel insights into the biology of 14-3-3ζ pseudogenes, emphasizing their possible functional roles and associations with human diseases. Enhancements in figures’ resolution or labeling should be considered and might be beneficial to further improve clarity and impact. Overall, this short report through bioinformatics analysis is interesting and useful for further medical applications. Accordingly, this reviewer recommends publication.
Author Response
We thank all the reviewers for their time and efforts. We are happy to see that all reviewers found our study foundational, novel, and logical. Below, we provide point-by-point responses to all the comments raised by reviewers.
Reviewer#1
In the manuscript "Structural Features and Physiological Associations of Human 14-3-3ζ Pseudogenes", Lughmani et al investigated the characteristics and potential physiological roles of YWHAZ pseudogenes in the human genome. The 14-3-3ζ is a highly conserved and universally present protein, and its adaptor function allows it to interact with hundreds of proteins, participate in several signaling pathways, and regulate cytokine signaling and inflammatory diseases, including rheumatoid arthritis, cancer, and neurological disorders. Although lacking experimental data, this manuscript provided a bioinformatics analysis of these pseudogenes' structure, sequence conservation, and their possible links to various human diseases. The study employed bioinformatics tools to analyze pseudogene sequences and their expression across different tissues, suggesting that these pseudogenes may play significant roles in physiological processes. The manuscript is well-structured and presents the information in a logical sequence.
We are encouraged by the reviewer’s opinion of our study as logical and well-structured. We thank you.
Minor grammatical or typographical errors could be addressed for clarity. In addition, this manuscript is not written as in mdpi format, including using different references styles. Figures and Tables were not structured.
Thank you for pointing it out. We have modified the manuscript to meet the journal's requirements with improved figures, tables, and references.
The methods used for genetic feature analysis, sequence alignment, and structure simulation were detailed and appropriate for the study's aims. However, it would benefit to explain how these methodologies can be associated to contribute to the novelty and conclusions of the study. Indeed, this study provided novel insights into the biology of 14-3-3ζ pseudogenes, emphasizing their possible functional roles and associations with human diseases. Enhancements in figures’ resolution or labeling should be considered and might be beneficial to further improve clarity and impact. Overall, this short report through bioinformatics analysis is interesting and useful for further medical applications. Accordingly, this reviewer recommends publication.
We are happy to see that the reviewer finds this study useful and shares our excitement about its potential use in future medical applications. This is a comprehensive bioinformatic study of 14-3-3z pseudogenes, in which we identified new information about their existence and putative associations with human diseases using standard tools. We agree with the reviewer that better resolution and labeling can help further improve impact. Therefore, we provide higher-resolution figures with clear and consistent labels, particularly for Fig 1 and Fig 4.

Reviewer 2 Report
Comments and Suggestions for Authors
The YWHAZ pseudogenes exhibit significant sequence and structural conservation with the parent protein, 14-3-3ζ, suggesting potential functional similarities. This conservation implies that pseudogenes may retain key motifs and domains important for protein interactions and cellular functions. The high homology in amino acid sequences indicates a potential for similar functional capabilities between pseudogenes and the parent protein. Structural resemblance between pseudogenes and 14-3-3ζ suggests a possible role in supporting physiological functions. Understanding the impact of these structural similarities is essential for elucidating the functional implications of YWHAZ pseudogenes in cellular processes and disease pathways.
The value and importance of this paper lie in its contribution to advancing our understanding of pseudogene biology, gene regulation, and disease mechanisms, with implications for future research, clinical applications, and therapeutic developments in the field of molecular biology and genetics.
However, the limitations of this paper are as follows.
1. The study primarily focuses on the structural and sequence conservation of YWHAZ pseudogenes and their potential implications. However, the functional validation of these pseudogenes and their specific roles in cellular processes or disease pathways is not extensively explored. Further experimental studies are needed to validate the functional relevance of these pseudogenes.
2. The analysis of YWHAZ pseudogenes and their expression profiles may be limited by the sample size and diversity of the datasets used. A larger and more diverse dataset encompassing a wider range of tissues, cell types, and disease conditions could provide a more comprehensive understanding of the functional significance of these pseudogenes.
3. The interpretation of gene expression data and the structural conservation of pseudogenes may be subject to biases or limitations inherent in bioinformatics tools and algorithms used for analysis. Different methodologies and software tools can lead to varying results, affecting the conclusions drawn from the data.
4. While the paper discusses the potential implications of YWHAZ pseudogenes in musculoskeletal and neurological disorders, the direct clinical relevance of these pseudogenes in disease diagnosis, prognosis, or treatment strategies is not fully elucidated. Further studies linking pseudogene expression to clinical outcomes are needed to establish their clinical significance.
5. The study highlights the tissue-specific expression patterns of YWHAZ pseudogenes in certain cell types and tissues. However, the detailed mechanisms underlying the tissue-specific regulation of these pseudogenes and their functional consequences remain to be fully understood.
6. There may be a potential bias towards reporting significant findings or results supporting the hypotheses, leading to an incomplete representation of the data. Addressing publication bias and ensuring transparency in reporting all results, including negative findings, can enhance the credibility of the study.
7. The paper suggests future research directions and the need for further investigation into the regulatory roles of pseudogenes in gene expression and disease pathways. While the study lays the groundwork for understanding YWHAZ pseudogenes, additional research is required to uncover their full functional significance.
Author Response
We thank all the reviewers for their time and efforts. We are happy to see that all reviewers found our study foundational, novel, and logical. Below, we provide point-by-point responses to all the comments raised by reviewers.
Reviewer#2
The YWHAZ pseudogenes exhibit significant sequence and structural conservation with the parent protein, 14-3-3ζ, suggesting potential functional similarities. This conservation implies that pseudogenes may retain key motifs and domains important for protein interactions and cellular functions. The high homology in amino acid sequences indicates a potential for similar functional capabilities between pseudogenes and the parent protein. Structural resemblance between pseudogenes and 14-3-3ζ suggests a possible role in supporting physiological functions. Understanding the impact of these structural similarities is essential for elucidating the functional implications of YWHAZ pseudogenes in cellular processes and disease pathways.
The value and importance of this paper lie in its contribution to advancing our understanding of pseudogene biology, gene regulation, and disease mechanisms, with implications for future research, clinical applications, and therapeutic developments in the field of molecular biology and genetics.
We thank the reviewer for identifying the strength of our manuscript. Even if we do not have answers to every question, this study will attract attention to YWHAZ pseudogenes and be foundational for future studies.
However, the limitations of this paper are as follows.
- The study primarily focuses on the structural and sequence conservation of YWHAZ pseudogenes and their potential implications. However, the functional validation of these pseudogenes and their specific roles in cellular processes or disease pathways is not extensively explored. Further experimental studies are needed to validate the functional relevance of these pseudogenes.
This is a bioinformatics-based study. We understand that experimental studies are needed to claim their functional relevance. A couple of studies where pseudogenes were functionally tested had only a limited number of pseudogenes. In our case, it won't be straightforward due to multiple genomic locations. Undertaking this study without impacting other genes will be a substantial technical and scientific challenge. We need an advanced understanding of pseudogenes and neighbors before undertaking that study. Therefore, we performed bioinformatic analyses only and carefully claimed putative associations throughout our study.
- The analysis of YWHAZ pseudogenes and their expression profiles may be limited by the sample size and diversity of the datasets used. A larger and more diverse dataset encompassing a wider range of tissues, cell types, and disease conditions could provide a more comprehensive understanding of the functional significance of these pseudogenes.
We have used several relevant portals, including GTEx, E-MTAB, OMIM, DECIPHER, and ClinVar, to evaluate the expression of pseudogenes. We used data from over 50 tissues for comparison (Fig 4). However, limited data are available for the cell types in the dataset.
- The interpretation of gene expression data and the structural conservation of pseudogenes may be subject to biases or limitations inherent in bioinformatics tools and algorithms used for analysis. Different methodologies and software tools can lead to varying results, affecting the conclusions drawn from the data.
As all algorithms are designed with certain assumptions and will have an inherent basis, it has been documented that any bioinformatics-based analysis will have a particular bias (PMID:26232237). However, we can ensure that, as responsible researchers, we report findings as we obtain from various datasets. Our findings cover multiple diseases including neurological, autoimmune, and cancer.
- While the paper discusses the potential implications of YWHAZ pseudogenes in musculoskeletal and neurological disorders, the direct clinical relevance of these pseudogenes in disease diagnosis, prognosis, or treatment strategies is not fully elucidated. Further studies linking pseudogene expression to clinical outcomes are needed to establish their clinical significance.
We agree with the reviewer that results derived from dataset analysis can only indicate associations with human diseases. To gain a broad understanding and validate results, we used multiple data sets- OMIM, ClinVar, and DECIPHER to identify human relevance. To strengthen these results further, we used literature mining and found several examples that further supported the association of pseudogenes and human disease. This includes the presence of YWHAZ pseudogenes in human psoriatic arthritis, glioblastoma, and several cancers (line numbers 310-335).
- The study highlights the tissue-specific expression patterns of YWHAZ pseudogenes in certain cell types and tissues. However, the detailed mechanisms underlying the tissue-specific regulation of these pseudogenes and their functional consequences remain to be fully understood.
Pseudogenes are a relatively understudied field in biology. Our is the first bioinformatic assessment describing YWHAZ pseudogenes in the human genome. Their regulations and mechanisms are beyond the scope of this study and will be examined in the future.
- There may be a potential bias towards reporting significant findings or results supporting the hypotheses, leading to an incomplete representation of the data. Addressing publication bias and ensuring transparency in reporting all results, including negative findings, can enhance the credibility of the study.
As we mentioned above, algorithms have an inherent bias, but the results shown here are accurate to what has been obtained from the investigation.
- The paper suggests future research directions and the need for further investigation into the regulatory roles of pseudogenes in gene expression and disease pathways. While the study lays the groundwork for understanding YWHAZ pseudogenes, additional research is required to uncover their full functional significance.
We are happy that the reviewer agrees that this is a foundational study and opens many avenues for future studies.

Reviewer 3 Report
Comments and Suggestions for Authors
The authors present a very interesting research on the plausible functionalities of 14-3-3ζ Pseudogenes, previously not investigated.
The field of genetic regulation of WT and pseudogenes and their roles in physiology and pathophysiology is emerging.
Although data is scarce, the authors are advised to perhaps investigate certain downstream analyses to underpin certain pathophysiological mechanisms these pseudogenes are involved in, as OMIM, ClinVar and Decipher data suggests they are likely involved in immune regulation and functioning.
Author Response
We thank all the reviewers for their time and efforts. We are happy to see that all reviewers found our study foundational, novel, and logical. Below, we provide point-by-point responses to all the comments raised by reviewers.
Reviewer#3
The authors present a very interesting research on the plausible functionalities of 14-3-3ζ Pseudogenes, previously not investigated.
We thank the reviewer for encouraging comments.
The field of genetic regulation of WT and pseudogenes and their roles in physiology and pathophysiology is emerging.
We agree with the reviewer.
Although data is scarce, the authors are advised to perhaps investigate certain downstream analyses to underpin certain pathophysiological mechanisms these pseudogenes are involved in, as OMIM, ClinVar and Decipher data suggests they are likely involved in immune regulation and functioning.
We appreciate that the reviewer recognizes that limited data are available on these pseudogenes. However, at your suggestion, we performed further analysis and found that both gains and losses of pseudogene can have a significant clinical phenotype. This will suggest that a regulatory mechanism must exist for their expression, which further demands that it be studied. We added this to the revised draft (Lines 303-306).

Round 2
Reviewer 3 Report
Comments and Suggestions for Authors
The authors have provided suffcient data on the suggestion to perform downstream analyses to investigate potential pathophysiological mechanisms the WT and pseudogenes might be involved in.